# Measuring childhood maltreatment: Psychometric properties of the Norwegian version of the Maltreatment and Abuse Chronology of Exposure (MACE) scale

**Roar Fosse**[1]*, **Dag Vegard Skjelstad**[1,2], **Inga Schalinski**[3], **Dorothea Thekkumthala**[4], **Thomas Elbert**[4], **Chris Margaret Aanondsen**[5,6], **Hanne Klæboe Greger**[5,6], **Thomas Jozefiak**[5]

**1** Department of Mental health and addiction, Vestre Viken hospital trust, Asker, Norway, **2** Department of Psychology, University of Oslo, Oslo, Norway, **3** Charité–Universitätsmedizin Berlin, corporate member of Freie Universität Berlin, Humboldt-Universität zu Berlin, and Berlin Institute of Health (BIH), Institute of Medical Psychology, Berlin, Germany, **4** Department of Psychology, University of Konstanz, Konstanz, Germany, **5** Institute of Mental Health, Faculty of Medicine and Health Sciences, Norwegian University of Science and Technology, Trondheim, Norway, **6** Department of Child and Adolescent Psychiatry, St. Olavs University hospital, Trondheim, Norway

\* roar.fosse@vestreviken.no

## Abstract

### Purpose

Adverse childhood experiences in sensitive periods of the developing brain render the individual at a life-long risk for a broad spectrum of aberrant health outcomes. However, there is a lack of scales for the comprehensive assessment of adverse childhood experiences providing information of various types and the age of occurrence. Based on the complete, experimental version of the Maltreatment and abuse chronology of exposure (MACE-X) scale, the present study aimed to develop and psychometrically test a Norwegian version of MACE.

### Methods

The 75-item MACE-X was translated from German to Norwegian and administered as a self-report measure to 90 outpatients and 145 employees at a Division of specialized mental health care in South-Eastern Norway. The outpatients also completed the Childhood trauma questionnaire (CTQ) and the Symptom checklist 90 (SCL-90) to investigate convergent and predictive validity. To investigate test-retest reliability, outpatients completed MACE once more two weeks later.

### Results

Rasch analysis and Anderson likelihood ratio tests on the combined outpatient and employee data resulted in a 55 item version of the Norwegian MACE. In the outpatient group, test-retest reliability of the MACE-55 was excellent for total scores (ICC $\geq$ 0.94) and

**Data Availability Statement:** All relevant data are within the paper and its Supporting Information files.

**Funding:** The Regional center for child and youth mental health and child welfare at the Norwegian University of Science and Technology funded the translation of the Childhood Trauma Questionnaire into Norwegian. The funder had no role in study design, data collection and analysis, decision to publish, or preparation of the manuscript.

**Competing interests:** The authors have declared that no competing interests exist.

good to excellent for 10 subscale scores (ICC $\geq$ 0.82). Convergent validity with the CTQ was moderate to high for both total scores (0.63 $\geq$ r $\geq$ 0.86) and subscale scores (0.56 $\geq$ r $\geq$ 0.82). As compared to CTQ total scores, a MACE total score that combined severity and duration of exposure was numerically more strongly associated with overall psychiatric symptoms and each of nine symptom domains on the SCL-90.

## Conclusions

The newly developed Norwegian MACE comprehensively assesses past exposure to adverse childhood experiences with high psychometric properties. This scale is a useful tool for research questions addressing sensitive periods for childhood adversities and associated health phenotypes.

## Introduction

Since the seminal work of Felliti, Anda and colleagues [1] it is increasingly acknowledged that adverse childhood experiences are central risk factors for a broad spectrum of mental disorders including mood disorders, suicidality, post-traumatic stress disorder, personality disorders, substance use disorder, and psychosis [2–5]. This multitude has stimulated research on psychological and biological trajectories of adverse childhood experiences. Common sequels include increased sensitivity to stress, diminished self-esteem, aberrant cognitive functioning, dissociation, and interpersonal problems. Increasing amounts of research report that the psychological changes often are accompanied by a matrix of alterations in brain structure and function. These include volume reductions in the prefrontal cortex and hippocampus, with atrophy of excitatory apical dendrites on pyramidal cells and reduced number and activity in parvalbumin-containing interneurons, and a changed developmental course of the amygdala. The consequence is altered neural network activity that includes compromised frontal cortical control over information processing [6–8]. This matrix of changes is also seen in mental disorders, where they are closely associated with exposure to adverse childhood experiences [9, 10]. Evidence continues to accumulate that effects of childhood maltreatment are not circumscribed to mental disorders but increase the risk of an array of somatic disorders as well (see for example [11, 12]).

Existing scales to assess adverse childhood experiences, including the gold standard instruments Childhood Trauma Questionnaire (CTQ) [13, 14] and the Adverse Childhood Experience scale (ACE) [1], have several limitations. One limitation is a narrow range of childhood adversities covered, such as the exclusion of bullying in both the CTQ and ACE. Further examples are the exclusion of witnessing domestic violence in the CTQ and restricting witnessing domestic violence to mothers and stepmothers only in the ACE, and excluding witnessing such behavior toward e.g. siblings [15]. Another limitation is the low number of items, such as only 10 in ACE, which makes them easier to apply in many circumstances but reduces the precision of individual predictions that can be made. In addition, neither the CTQ nor the ACE includes information on timing and duration of exposure or of how exposure levels change across development. Hence, these instruments cannot be used to determine variable effects of adverse childhood experiences as a function of temporal overlap with sensitive developmental periods, which may be central to their degree of impact [15].

In order to improve upon the measurement of adverse childhood experiences, researchers have developed the Maltreatment and Abuse Chronology of Exposure (MACE) scale [15–17].

The original, complete experimental version of MACE (MACE-X) consists of 75 questions to retrospectively assess exposure to 10 types of adverse childhood experiences in adults, where each item is considered for each year of development until age 18. Studies found very good psychometric properties of US and German versions of MACE. Moreover, MACE has been indicated to account for substantially more of the variance in psychiatric symptom ratings compared to the CTQ and ACE [15]. US and German versions of the instrument have already been successfully employed in studies of psychiatric symptoms and a variety of other sequels of adverse childhood experiences, including sleep disruption, stress sensitivity and resilience, and altered brain functioning and structure [18–23].

In this study, our overarching aim was to develop a Norwegian MACE and to investigate its psychometric properties. Specific study aims were to:

i. Create the Norwegian MACE-X based on the German 75-item MACE-X version using the back-translation procedure

ii. Extract the most suitable items from the translated MACE-X to develop a Norwegian version of MACE with suitable scales to measure 10 different forms of adversity

iii. Assess test-retest reliability of the resulting, Norwegian MACE

iv. Assess convergent validity between the Norwegian MACE and the gold standard CTQ

v. Estimate relative predictive validity of the Norwegian MACE compared to CTQ for patients' psychiatric symptoms

## Materials and methods

### Setting and ethics

We administered the Norwegian MACE-X as a self-report measure to two groups of study participants. First, a group of employees at the Division of Mental Health and Addiction, Vestre Viken hospital trust in Norway completed the MACE once. Second, adult outpatients at the same Division at Vestre Viken completed the MACE-X as well as the CTQ and the symptom measure Symptom Check List, 90 item version (SCL-90). Two weeks after the first assessment, the outpatients completed the MACE-X once more to assess test-retest reliability. Vestre Viken hospital trust provides specialized mental health services for a community population of approximately 500,000.

The Regional Committee for Research Ethics in Health region south-east, Norway, considered the study to fall outside their mandate. Since all study data were anonymous, the Data Protection Service for Research at Vestre Viken concluded that the study did not require any application of approval. All participants were given written information about the study and consented to participate. The study was conducted completely anonymously.

### Participants

Hundred and forty-five Norwegian-speaking adults above 18 years were recruited among employees at Division of Mental health and Addiction (120 females; 83%), mean age 47.0 years (range 22–67, SD 10.8). This group was recruited through an email sent from the Division Director, with information about the study and a link to complete MACE-X in the Confirmit poll system–an online survey technology.

Twenty-three clinicians providing outpatient treatment at two District psychiatric centers in Vestre Viken recruited eligible patients to the study during their therapy sessions. Inclusion

criteria were 18–50 years of age, mastery of Norwegian language, and sufficient cognitive ability and mental health to complete the study instruments. Using pen and paper, consenting patients completed MACE-X, CTQ and SCL-90 prior to their next appointment with their therapist, and the MACE-X a second time prior to an appointment two weeks later. Ninety participants were included in this group (62 females; 69%), mean age 34.6 years (SD = 10.6). Primary diagnosis (ICD-10) as provided by the clinicians, were depression (F32-34, n = 33), PTSD (F43.1, n = 14), phobic and anxiety disorders (F40-41, n = 14), personality disorders (F60-69, n = 14), hyperkinetic disorder (F90, n = 5), alcohol or drug abuse (F10-19, n = 4), and 6 other diagnoses.

## Measures

The complete, experimental Maltreatment and Abuse Chronology of Exposure (MACE-X) scale uses 75 questions to assess exposure to the following 10 types of maltreatment before age 18: Parental verbal abuse (PVA), Parental non-verbal emotional abuse (PNEVA), Parental physical maltreatment (PPA), Emotional neglect (EN), Physical neglect (PN), Witnessing interpersonal violence to parents (WITP), Witnessing violence to siblings (WITS), Peer verbal/ emotional abuse (PEERE), Peer physical bullying (PEERP), and Sexual abuse (familial and extra-familial; SEXA)[15, 16]. Each item is scored as yes or no and for each year of exposure from age 1 to 18. Eight of the items ask for positive experiences, and responses are reversed before analysis (see Table 1). MACE has been reported to exhibit adequate test-retest reliability for both total scores and subscale scores, and to have excellent convergent and predictive validity [15, 16].

We used the following summated MACE scores in statistical analysis:

i. Ten Subscale scores, summing up items with positive responses for each subscale, with each subscale score fitted into a 0 to 10 point scale

ii. A total Neglect score–the sum of the two subscales for emotional neglect and physical neglect (range 0–20), and a total Abuse score–the sum of the other eight subscales (range 0–80)

iii. An overall Sum score which is the sum of the scores on the 10 subscales (range 0–100)

iv. A Multiplicity score for the number of different types of maltreatments (range 0–10), where each type is required to be reported above a clinical cutoff level as calculated with reference to clinical cut-offs on CTQ (see also Statistics)

v. A total Duration score, which is the number of years with exposure to at least one type of maltreatment above clinical cut-off as calculated with reference to clinical cut-offs on CTQ (range 0 to 18)

**Table 1. Item fit statistics for the subscale Parental Verbal Abuse/ PVA.**

| Item | item difficulty β (SE) | Outfit MSQ | Infit MSQ |
|---|---|---|---|
| 1. Swore at you, called you names, insulted | 0.77 (0.20) | 0.68 | 0.77 |
| 2. Said hurtful things, made you feel humiliated | -0.85 (0.20) | 0.86 | 0.95 |
| 3. Yelled or screamed at you | -0.29 (0.19) | 1.04 | 1.04 |
| 4. Acted in a way that made you afraid that you might be physically hurt | 0.36 (0.20) | 1.04 | 1.07 |

$X^2(3) = 6.56$, p = .088 (median of age as the split criterion)
$X^2(3) = 4.24$, p = .237 (gender as the split criterion)

vi. A total Sum-by-Duration score, derived by first making a sum score for each age level (1–18) by adding up scores for the 10 subscales at that age, second, adding these scores for the 18 age levels, and, third, dividing by 18 to obtain a 0–100 range scale. That is, here we use the approach outlined for the MACE Sum score in point (iii) above separately for each age level, then add together these age-level sum scores for the 18 age levels, and finally divide by 18

To develop a Norwegian MACE, two experienced bilingual (German/Norwegian) professionals working in the mental health field, independently translated the German MACE-X ("Belastende Kindheitserfahrungen"–KERF) [16] to Norwegian. Together with an additional bilingual professional, they developed a consensus translation taking into account semantic, conceptual, lexical, and cultural differences. A professional translator then back-translated the consensus version to German, which in turn was approved by researchers of the German MACE group. Finally, 10 male and female adult respondents (half of them patients) provided feedback on words they did not understand and unacceptable expressions in the approved Norwegian translation, leading to some minor amendments.

The Childhood Trauma Questionnaire (CTQ) [13, 24] is widely used to measure childhood maltreatment. The CTQ uses five items each to assess five types of maltreatment: Emotional neglect, Physical neglect, Emotional abuse, Sexual abuse, and Physical abuse. Each item is scored on a five-point Likert-scale (ranging from 'never true' to 'very often true'). The instrument includes a 3-item minimization scale (e.g., "best family in the world") which gives an estimate of response validity (the tendency to idealize one's family). A total Severity score is obtained by summing the scores for the 25 items (three minimization items excluded). Subscale scores are created in the same manner. In addition, a Multiplicity score is generated based on scores above cutoff on the five subscales, as proposed by Bernstein and Finke [24].

A previous Norwegian translation of the CTQ developed for research purposes showed acceptable psychometric properties, with good reliability and satisfactory accuracy [25]. For the present study, a new Norwegian translation was required because of NCS Pearson's copyrights to the CTQ. The translation was carried out according to international standards [26].

Symptom Checklist 90 (SCL-90) is a widely applied self-assessment questionnaire with 90 items for nine domains of current mental health problems: Somatization, Obsessive-compulsive, Interpersonal sensitivity, Depression, Anxiety, Hostility, Phobic anxiety, Paranoid ideation, and Psychoticism [27, 28]. The nine domains contain from 6 to 13 items, each rated for degree of distress on a five point Likert scale ('not at all' = 0 to 'extremely' = 4). We calculated: the General Severity Index (GSI), which is the mean score of all items, the Positive symptom total (PST), which is the number of items scored above zero; and severity scores for each symptom domain calculated in the same manner as the GSI. The Norwegian version of SCL-90 [29, 30] is used as a standard assessment tool in many clinical settings, including Vestre Viken.

## Statistical analysis

**Item to subscale assignment and scaling.** For each of the 10 MACE subscales, we initially considered all items previously found to be included in the subscale in either the German or American version [15, 16]. We determined a simple Rasch model for each subscale, aiming to include at least 4 items that measure the latent trait (see [15]). The different items in a subscale will have varying levels of difficulty, which will reflect the latent trait dimension. Using "Physical abuse" as example, individuals will report "intentionally pushed, grabbed, shoved, slapped" (item 7) more often than "hit you so hard that it left marks for more than a few minutes" (item 8), and this item in turn less often than "hit you so hard that you received medical attention" (item 9). Individuals with the highest exposure to physical abuse are expected to report the

most difficult item as well as endorsing the easier items, while individuals with low levels are likely to report the "easiest" item. We used the libraries extended Rasch Modelling (eRM) [31] and latent trait models (ltm) [32] in R version 3.6.0 [33] to calculate simple Rasch models for each subscale and for plots. First, appropriate items were identified based on the mean square fit criteria (X2/df) as measured by infit and outfit mean square fits. Infit depicts unexpected responses to the item with a difficulty close to the individual exposure levels, while outfit square measures unexpected responses to items with a distinct difficulty to the individual exposure level. The thresholds of acceptable mean square fits are subject to an ongoing debate. While there is an agreement that mean square fits should not exceed 1.5, and not be lower than 0.5, Linacre, and colleagues [34] applied a maximum of 1.3 and a minimum of 0.7 for mean square fits for their data. For the present analysis, acceptable fits were considered as those falling into the range of 0.7 to 1.3. Fits that fell into the range of 0.5 to 0.7 as well as 1.3 to 1.5 were considered as "sufficient", however, they were critically inspected and evaluated and only included for content purposes, or to obtain at least 4 items per subscale. For the modeling of each subscale, we first evaluated the outfit before the infit. Furthermore, high mean square fits were considered more seriously because they threaten overall fit more (modeling random-ness) than low square fits (indicating too predictable values). Therefore, we considered all mean square values above 1.5 as "misfits" and excluded them rigorously from the subscale. Items with mean square fits below 0.5 were kept if the content appeared to be important for the respective subscale. Data from all participants (both patients and employees) were used in these analyses.

Following the optimization based on mean square fits on the item level, we performed global tests using the Andersen likelihood ratio (LR) test [35] with age (participant median) and gender as split criteria. This is a powerful test to detect differential item functioning that may occur if younger and older individuals (or men and women) at the same exposure level would report items with a different probability. When the Andersen LR test was significant, we used the Wald test to detect items with differential item parameters. For each subscale, we plotted the test information function and estimated the test's precision along the whole range of exposure levels. For each scale, we aimed to detect individuals with clinically significant exposure levels, based on the optimal discrimination between moderate to higher level of exposure. Therefore, we aimed to obtain the peak of the overall test information function yielded by the subscale around logit scores 0–2.

**Test-retest reliability.** In all subsequent tests, we analyzed the Norwegian version of MACE resulting from the above analysis. First, we assessed test-retest reliability for MACE total scores and the 10 subscale scores using 2-way mixed effects ANOVA (absolute agreement definition) for intraclass correlation coefficients based on a test-retest interval of two weeks (ICC) [36].

**Convergent validity.** Comparing the Norwegian MACE with CTQ, we first estimated convergent validity using Pearson correlation coefficients for total scores and subscale scores, with the expectation that convergent validity would manifest as correlation coefficients > 0.6. Here, we correlated the CTQ total scores with two versions of total scores on MACE; one that included all MACE subscales and one where we excluded four subscales that are not covered by the CTQ (Peer verbal/ emotional abuse, Peer physical bullying, Witnessing interpersonal violence to parents, and Witnessing interpersonal violence to siblings). Second, we estimated the ability of MACE subscales to predict the absence and presence of clinically relevant expo-sure levels on corresponding CTQ subscales. To define clinically relevant exposure levels we used the cutoff scores proposed by Bernstein and Finke [24] for moderate to severe levels of exposure for the five CTQ subscales. We used ROC-analysis (receiver operating characteris-tics) to determine the discrimination properties of the raw scores on the matching MACE

subscales (Parental verbal abuse, Emotional neglect, Physical neglect, Parental-non-verbal emotional abuse, Parental physical abuse, and Sexual abuse). Parameters of diagnostic accuracy (sensitivity and specificity) were retrieved to determine the optimal cutoff value of the raw score on the subscales. The aim of the cutoff was to identify as many cases as possible with clinically relevant exposure levels; thus, sensitivity was evaluated first (with at least 0.70) and then specificity. Optimal tradeoff was determined as the highest sensitivity–specificity combination. There was no external reference for the absence and presence of the MACE subscales emotional and physical abuse by peers or for witnessed violence towards siblings and parents. For these subscales, the cutoff of the German MACE was adopted and corrected for the number of items per subscale.

**Predictive validity.** We first calculated Pearson correlations between total scores on the Norwegian MACE (Sum, Multiplicity, and Sum-by-Duration) and the CTQ (Sum, Multiplicity), respectively, and total scores and subscale scores on the SCL-90. To compare the size of the correlations with SCL-90 for MACE and CTQ, we used Hotelling's t-test for within sample comparisons.

Analyses were performed in R version 3.6.0 [33] and SPSS version 23 [37].

## Results

### MACE Item to subscale assignment and scaling

Below, we present results separately for each of the 10 MACE subscales, based on the combined patient and employee group (n = 235). See Figures A-J in S1 File with additional details.

**Subscale 1. Parental Verbal Abuse (PVA).** We considered five MACE items for inclusion in PVA subscale (items 1–5). Item 5 was removed due to a misfit of the outfit score of 1.57. In the final scale version, all mean square fits were below 1.3, even though item 1 obtained a somewhat small outfit of 0.68 indicating some degree of item redundancy. The Andersen test was not significant, indicating fit to the Rasch model. The test information function showed that the highest information of the scale was within the target range of logit scores between 0–2, indicating optimal discrimination between moderate to higher level of exposure with 45.3% of the total information (Table 1).

**Subscale 2. Emotional Neglect (EN).** Ten MACE items were initially included in the EN subscale, however, two items (items 53 and 54) demonstrated high outfits (> 1.41). Because of the potential ambiguous content of these items (e.g. "caregivers are emotionally unavailable for good reasons"), we decided to remove both items from the scale. MACE items 56 and 73 showed a very similar difficulty; we dropped item 73 because it also appeared suitable for the subscale physical neglect (as in the American MACE). The final scale consisted of 7 items with mean square fits ranging from 0.61 to 1.38. Item 52, "the father being unavailable for poor reasons", had a critical high outfit. However, we kept the item in order to equally consider caregivers in the scale. While item 75 had a low outfit of 0.61, indicating some degree of item redundancy, we retained this item since it also is retained in the German MACE. Despite this, the overall scale showed an appropriate fit for younger and older individuals and for men and women. The test information curve had the highest information within the target range 0–2, indicating optimal discrimination between moderate to higher level of exposure with 43.5% of the information (Table 2).

**Subscale 3. Physical Neglect (PN).** The PN subscale consists of 5 items in the German MACE. We also added MACE item 73 to the item pool, in accordance with the American MACE. Mean square fits ranged from 0.61 to 1.14. One item (item 59) had a low outfit (< 0.70) indicating some degree of overfit. We decided to keep item 59 because its removal affected the other items' mean square fits. The Andersen test was significant when comparing

**Table 2.  Item fit statistics for the subscale Emotional Neglect/ EN.**

$X^2(6) = = 5.06, p = .536$ (median of age as the split criterion)
$X^2(6) = 2.48, p = .871$ (gender as the split criterion)

| Item | item difficulty β (SE) | Outfit MSQ | Infit MSQ |
|---|---|---|---|
| 51. Mother unavailable for poor reasons | 0.96 (0.18) | 0.99 | 1.02 |
| 52. Father unavailable for poor reasons | 0.28 (0.17) | 1.38 | 1.21 |
| 56. A parent or other important parental figure did not have the time or interest to talk to you | -0.01 (0.17) | 1.24 | 1.20 |
| 57. One or more individuals of your family made you feel loved. | 0.57 (0.17) | 0.78 | 0.88 |
| 58. One or more individuals in your family helped you feel important or special | -0.42 (0.16) | 1.01 | 1.04 |
| 74. People in your family felt close to each other | -0.54 (0.17) | 0.72 | 0.77 |
| 75. Your family was a source of strength and support. | -0.83 (0.17) | 0.61 | 0.73 |

the model for men and women but not with age as a split criterion. The Wald test indicated invariance for men and women on item 63 ($Z = 2.13$, $p = .033$) and item 73r ($Z = -2.89$, $p = .004$). The peak of the test information curve was within the target range for the overall sample, however, it showed a difference for men and women (Table 3).

**Subscale 4. Parental-Nonverbal-Emotional Abuse (PNVEA).** The German MACE included 5 items for this subscale. Since one of these, item 4, was added to the subscale PVA, it was removed from the item pool for PNVEA. We added MACE item 66 to the item pool following the American MACE. All mean square fits were in the acceptable range (0.70 to 0.98). The overall scale did not deviate from the Rasch model when splitting for younger and older individuals or for men and women. The maximum of the test information curve revealed an optimal discrimination for moderate to severe exposure levels with 39.3% of the information between logit scores 0–2 (Table 4).

**Subscale 5. Parental Physical Abuse (PPA).** We considered MACE items 7 to 12 for inclusion in the subscale. While none of these items had high mean square fits > 1.3, some had low outfits (items 8 and 9). None of the items deviated from the one-dimensional Rasch model, but we could not fit the model for younger and older individuals. The removal of items with low mean square fits did help fitting the Rasch model for younger and older individuals. Therefore, we chose to include all items in the final scale. The Wald test indicated invariance for younger and older individuals on item 7 ($Z = 3.39$, $p = .001$), item 10 ($Z = 2.12$, $p = .034$),

**Table 3.  Item fit statistics for the subscale Physical Neglect/ PN.**

$X^2(5) = 4.84, df = 5, p = .436$ (median of age as the split criterion)
$X^2(5) = 12.72, df = 5, p = .026$ (gender as the split criterion)

| Item | item difficulty β (SE) | Outfit MSQ | Infit MSQ |
|---|---|---|---|
| 59. One or more family members were there to take care of you and protected you | -1.66 (0.21) | 0.61 | 0.76 |
| 60. One or more individuals in your family were there to take you to the doctor or ER if the need ever arose | -0.33 (0.21) | 0.85 | 0.86 |
| 62. You did not have enough to eat | 1.41 (0.28) | 1.14 | 0.78 |
| 63. You had to wear dirty clothes | 2.93 (0.45) | 0.81 | 0.79 |
| 64. You were left unsupervised at an age or in situations when you should have been supervised | -0.63 (0.21) | 1.05 | 1.08 |
| 73. People in your family looked out for each other | -1.73 (0.21) | 0.75 | 0.94 |

**Table 4. Item fit statistics for the subscale Parental-Nonverbal-Emotional Abuse/ PNVEA.**

$X^2(4) = 5.83$, p = .212 (median of age as the split criterion)
$X^2(4) = 0.77$, p = .942 (gender as the split criterion)

| Item | item difficulty β (SE) | Outfit MSQ | Infit MSQ |
|---|---|---|---|
| 6. Locked you in a closet, attic, basement or garage | 2.26 (0.30) | 0.981 | 0.701 |
| 55. A parent or other important parental caregiver was very difficult to please | -1.15 (0.17) | 0.885 | 0.978 |
| 65. You felt that you had to shoulder adult responsibilities | -0.93 (0.17) | 0.791 | 0.875 |
| 66. You felt that you family was under financial pressure. | 0.03 (0.18) | 0.948 | 0.975 |
| 67. One or more individuals kept important secrets or facts from you. | -.21 (0.17) | 0.983 | 0.97 |

and item 12 ($Z = -3.10$, $p = .002$). The overall test information showed most information between moderate to high exposure levels with 35.3% between logit scores 0–2 and 35.2% between logit scores 2–4 (Table 5).

**Subscale 6. Witnessing Violence Towards Parents (WITP).** Eight MACE items were considered for inclusion (items 31 to 38). Item 35, "adults living in the household hit your mother so hard that she received medical attention", had a low outfit = 0.23. We removed both this item and the corresponding item of the other parent (item 38). The final scale consisted of 6 items. Mean square fits were all below 1.3 and acceptable. All items had outfits > 0.5, while four items showed outfits < 0.7. We decided to keep these items because of the content of the scale. The Anderson test was not significant either with age or gender as a split criterion. The test information curve showed that the test best discriminated high exposure levels with logit scores between 2–4, including 36.1% of the information (Table 6).

**Subscale 7. Witnessing Violence Towards Siblings (WITS).** Initially, we included 6 MACE items (items 18 to 22 and 25) but removed item 20 due to a very low outfit = 0.23, leaving 5 items in the final version. While all outfit values were below 1.3, two items had a low outfit (0.48 for item 19 and 0.44 for item 22). We decided to keep item 22 because it assesses an important aspect of this scale. However, the Andersen tests were not significant when split for

**Table 5. Item fit statistics for the subscale Parental Physical Abuse/ PPA.**

$X^2(5) = 22.96$, p < .001 (median of age as the split criterion)
$X^2(5) = 5.12$, $p = .402$ (gender as the split criterion)

| Item | | item difficulty β (SE) | Outfit MSQ | Infit MSQ |
|---|---|---|---|---|
| 7. Intentionally pushed, pinched, slapped, etc. | | -1.76 (0.23) | 1.042 | 1.091 |
| 8. Hit you so hard that it left marks for more than a few minutes | | 0.54 (0.26) | 0.43 | 0.618 |
| 9. Hit you so hard, or intentionally harmed you in some way, that you received medical attention | | 2.62 (0.45) | 0.286 | 0.618 |
| 10. Spanked you with their open hand on you buttocks, arms or legs | | -1.13 (0.22) | 0.831 | 0.932 |
| 11. Spanked you on your bare buttocks | | -0.6 (0.23) | 0.797 | 0.854 |
| 12. Spanked you with an object e.g., strap, etc. | | 0.33 (0.25) | 1.112 | 1.21 |

**Table 6. Item fit statistics for the subscale Witnessing Violence Towards Parents/ WITP.**

$X^2(5) = 9.13$, $p = .104$ (median of age as the split criterion)
$X^2(5) = 9.12$, $p = .104$ (gender as the split criterion)

| Item | item difficulty β (SE) | Outfit MSQ | Infit MSQ |
|---|---|---|---|
| 31. Witnessed adults living in the household argue intensively with your mother | -1.68 (0.24) | 1.08 | 1.17 |
| 32. Witnessed adults living in the household argue intensively with your father | -2.07 (0.25) | 0.79 | 0.89 |
| 33. Saw adults living in household push, slap or throw something at your mother (stepmother, grandmother) | -0.25 (0.24) | 0.69 | 0.76 |
| 34. Saw adults hit mother (or other caregivers) so hard that it left marks for more than a few minutes | 0.81 (0.27) | 0.50 | 0.75 |
| 36. Saw adults living in household push, slap or throw something at your father | 0.72 (0.27) | 0.54 | 0.79 |
| 37. Saw adults hit father (stepfather, grandfather) so hard that it left marks for more than a few minutes | 2.45 (0.41) | 0.55 | 0.95 |

either age or gender. The peak of the test information curve indicated best discrimination for high exposure levels with 38.1% of the overall information for logit scores between 2 and 4 (Table 7).

**Subscale 8. Emotional Abuse by Peers (PEERE).** Initially, MACE items 39 to 43 were included in the item pool. However, item 43 was removed due to a high outfit (1.93), thus adding too much noise to the scale. All remaining items showed acceptable fits (ranging from 0.72 to 1.11). The Andersen test was not significant when splitting for younger and older individuals or for gender. The test information function showed that the highest information of the scale was within the target range 0–2, indicating optimal discrimination between moderate to higher level of exposure with 42.3% of the total information between logits scores 0–2 (Table 8).

**Subscale 9. Physical Abuse by Peers (PEERP).** Five items were considered for the subscale, items 44 to 48. Item 47 was removed due to very low outfit = 0.44. The outfit of the remaining items was below 1.3 and above 0.5. The Andersen test was not significant when splitting for age or for gender. The test information curve showed that the test was best at discriminating for higher exposure levels with logit scores between 2 and 4 (including 40.7% of the information (Table 9).

**Subscale 10. Sexual Abuse (SEXA).** From initially 9 included items, the final scale included 8 items since item 13 was removed due to a high outfit (1.44). The outfit of item 14

**Table 7. Item fit statistics for the subscale Witnessing Violence Towards Siblings/ WITS.**

$X^2(4) = 0.90$, $p = .924$ (median of age as the split criterion)
$X^2(4) = 4.52$, $p = .340$ (gender as the split criterion)

| Item | item difficulty β (SE) | Outfit MSQ | Infit MSQ |
|---|---|---|---|
| 18. Intentionally pushed, grabbed, shoved, slapped, pinched, punched, or kicked your sibling | -3.02 (0.44) | 1.19 | 0.66 |
| 19. Hit your sibling so hard that it left marks for more than a few minutes | -0.34 (0.33) | 0.48 | 0.62 |
| 21. Made inappropriate sexual comments | 1.05 (0.39) | 1.08 | 0.75 |
| 22. Touched or fondled your sibling in a sexual way | 1.89 (0.47) | 0.44 | 0.89 |
| 25. Threatened to harm your sibling | 0.42 (0.35) | 0.65 | 0.73 |

**Table 8. Item fit statistics for the subscale Emotional Abuse by Peers/ PEERE.**

$X^2(3) = 6.93$, $p = .074$ (median of age as the split criterion)
$X^2(3) = 2.02$, $p = .568$ (gender as the split criterion)

| Item | | item difficulty β(SE) | Outfit MSQ | Infit MSQ |
|---|---|---|---|---|
| 39. Swore, called you names/ insulted you | | 0.49 (0.19) | 0.94 | 0.98 |
| 40. Said hurtful things, made you feel humiliated | | -1.06 (0.21) | 0.72 | 0.82 |
| 41. Said things behind your back, spread rumors | | 0.08 (0.19) | 0.93 | 0.94 |
| 42. Excluded you from activities | | 0.49 (0.19) | 1.11 | 1.06 |

was somewhat high (1.38), however, in order to equally consider sexual abuse (with the same number of items) by parents as well as other adults, we decided to keep item 14 in the final scale. The Andersen test was not significant when using either age (item 17 was removed due to low frequency) or gender as a split criterion. The overall test information curve revealed the most information for high exposure levels with 41.8% of the overall information for logit scores between 2 to 4 (Table 10).

## Summary

The results of the item reduction analyses are summarized in Table 11. See also the resulting MACE-55 in S2 File.

## Test-retest reliability of Norwegian MACE

Eighty-seven outpatients had complete MACE-X forms at both baseline and follow up two weeks later. Using the MACE version with 55 items from the above analysis, test-retest reliability was excellent ($r \geq 0.90$) for total Sum scores and Multiplicity scores, for the total Neglect and total Abuse scores, and for two of the 10 subscales. Test-retest reliability was good ($r \geq 0.75$) for the remaining eight subscales (Table 12). For the MACE Duration score, 26 participants had incomplete MACE forms at follow up, having endorsed one or more items but failed to check off for any age levels of exposure. When excluding these 26 participants, test-retest reliability for the Duration score was good (n = 61, r = 0.84) for the remaining subgroup.

## Convergent Validity between MACE and CTQ

Convergent validity between MACE-55 and CTQ was tested in 87 participants in the outpatient group (three patients had incomplete/ missing CTQs). The overall MACE Sum score (mean = 50.5, SD = 17.9) and MACE Multiplicity score (mean = 1.7, SD = 1.6) correlated strongly with the

**Table 9. Item fit Statistics for the Subscale Physical Abuse by Peers/ PEERP.**

$X^2(3) = 3.01$, $p = .390$ (median of age as the split criterion)
$X^2(3) = 0.49$, $p = .922$ (gender as the split criterion)

| Item | item difficulty β (SE) | Outfit MSQ | Infit MSQ |
|---|---|---|---|
| 44. Threatened you to take your money or possessions | 1.47 (0.38) | 0.87 | 0.79 |
| 45. Forced or threatened you to do things | 0.15 (0.28) | 0.86 | 0.88 |
| 46. Intentionally pushed, grabbed, shoved, slapped, pinched, punched | -2.08 (0.30) | 0.57 | 0.71 |
| 48. Hit you so hard, or intentionally harmed you in some way, that you received or should have received medical attention | 0.46 (0.30) | 0.54 | 0.66 |

**Table 10. Item fit statistics for the subscale Sexual Abuse/ SEXA.**

$X^2(6) = 3.42$, $p = .754$ (median of age as the split criterion, item 17 had to be removed due to low frequency for the Andersen test)
$X^2(7) = 12.8$, $p = .077$ (gender as the split criterion)

| Item | item difficulty β (SE) | Outfit MSQ | Infit MSQ |
|---|---|---|---|
| 14. Parents touched or fondled your body in a sexual way | 0.37 (0.33) | 1.38 | 1.23 |
| 15. Parents had you touch their body in a sexual way | 0.96 (0.39) | 0.72 | 1.01 |
| 17. Attempted to have any type of sexual intercourse with you | 1.95 (0.55) | 0.51 | 0.74 |
| 27. Other adults touched or fondled you in a sexual way | -2.03 (0.27) | 1.11 | 1.23 |
| 28. Other adults had you touch their body in a sexual way | -0.36 (0.29) | 0.61 | 0.70 |
| 30. Other adults had sexual intercourse with you | -0.27 (0.29) | 0.72 | 0.72 |
| 49. Peer(s) forced you to engage in sexual activities against your will | -0.45 (0.29) | 0.90 | 0.97 |
| 50. Peer(s) forced you to do things sexually that you did not want to do. | -0.17 (0.30) | 0.74 | 0.90 |

CTQ Severity and Multiplicity scores (r's ≥ 0.78). All these correlations slightly increased when the MACE total score variables included only the six subscales that also are measured by the CTQ. The MACE duration score correlated less strongly with the two CTQ measures, consistent with this score measuring another aspect of childhood adversities (Table 13).

Moderate to strong correlations also were apparent between MACE subscale scores and the matching CTQ subscale scores (Table 14).

Table 15 shows results from ROC-analysis and analysis of sensitivity and specificity for the ability of five MACE subscales to predict the absence and presence of clinically relevant exposure levels on the concomitant CTQ subscales.

## Predictive validity

Regarding levels of explained variance of psychiatric symptoms, total scores on both MACE and CTQ were significantly correlated with total scores as well as most subscale scores on the

**Table 11. Distribution of 55 items in 10 MACE subscales after item reduction analyses.**

| Subscales | N items (item weight)* | Items (item numbers) in each subscale | |
|---|---|---|---|
| | | Item numbers in original MACE-75 | Corresponding item numbers in MACE-55 |
| 1. Parental verbal abuse (PVA) | 4 (2.5) | 1, 2, 3, 4 | 1, 2, 3, 4 |
| 2. Emotional neglect (EN) | 7 (1.43) | 51, 52, 56, 57r**, 58r, 74r, 75r | 39, 40, 42, 43r, 44r, 54r, 55r |
| 3. Physical neglect (PN) | 6 (1.67) | 59r, 60r, 62, 63, 64, 73r | 45r, 46r, 47, 48, 49, 53r |
| 4. Parental non-verbal emotional abuse (PNEVA) | 5 (2) | 6, 55, 65, 66, 67 | 5, 41, 50, 51, 52 |
| 5. Parental physical maltreatment (PPA) | 6 (1.67) | 7, 8, 9, 10, 11, 12 | 6, 7, 8, 9, 10, 11 |
| 6. Witnessed interpersonal violence to parents (WITP) | 6 (1.67) | 31, 32, 33, 34, 36, 37 | 23, 24, 25, 26, 27, 28 |
| 7. Witnessing violence to siblings (WITS) | 5 (2) | 18, 19, 21, 22, 25 | 15, 16, 17, 18, 19 |
| 8. Peer verbal abuse (PEERE) | 4 (2.5) | 39, 40, 41, 42 | 29, 30, 31, 32 |
| 9. Peer physical bullying (PEERP) | 4 (2.5) | 44, 45, 46, 48 | 33, 34, 35, 36 |
| 10. Childhood sexual abuse (SEXA) | 8 (1.25) | 14, 15, 17, 27, 28, 30, 49, 50 | 12, 13, 14, 20, 21, 22, 37, 38 |
| MACE total | 55 | 55 | 55 |

*Each of the 10 types of exposure is measured on a 0–10 scale, and item weight refers to how much (out of 10) that each item in the exposure subtype counts on its respective subscale, e.g. 10 /4 items = 2.5 weight per subscale item.

** Items marked with "r" were reversed prior to analysis.

**Table 12. Results of ICC test-retest reliability analysis for three MACE total scores and 10 forms of childhood adversities using absolute agreement, 2-way mixed effect ANOVA models (n = 87 outpatients).**

| Type of MACE score | Intraclass correlation | 95% CI | F-test | p-value |
|---|---|---|---|---|
| *Total scores* | | | | |
| Sum | 0.96 | [0.94–0.98] | 50.5 | $3.6 \times 10^{-50}$ |
| Multiplicity | 0.94 | [0.91–0.96] | 33.2 | $1.0 \times 10^{-42}$ |
| Duration[1] | 0.84 | [0.75–0.90] | 11.3 | $1.1 \times 10^{-17}$ |
| Abuse | 0.96 | [0.94–0.97] | 47.4 | $5.1 \times 10^{-49}$ |
| Neglect | 0.91 | [0.87–0.94] | 21.2 | $6.7 \times 10^{-35}$ |
| *Subscale scores* | | | | |
| Parental nonverbal emotional abuse | 0.88 | [0.82–0.92] | 15.0 | $3.9 \times 10^{-29}$ |
| Parental physical maltreatment | 0.95 | [0.92–0.97] | 38.1 | $3.9 \times 10^{-45}$ |
| Parental verbal abuse | 0.89 | [0.84–0.93] | 17.5 | $1.2 \times 10^{-31}$ |
| Peer verbal/ emotional abuse | 0.88 | [0.83–0.92] | 16.3 | $1.7 \times 10^{-30}$ |
| Peer physical bullying | 0.82 | [0.74–0.88] | 10.3 | $4.0 \times 10^{-23}$ |
| Emotional neglect | 0.88 | [0.82–0.92] | 15.8 | $5.5 \times 10^{-30}$ |
| Physical neglect | 0.88 | [0.82–0.92] | 15.8 | $6.2 \times 10^{-30}$ |
| Sexual abuse | 0.90 | [0.85–0.94] | 19.2 | $3.0 \times 10^{-33}$ |
| Witnessing interparental violence | 0.89 | [0.84–0.93] | 17.8 | $6.2 \times 10^{-32}$ |
| Witnessing violence to siblings | 0.94 | [0.91–0.96] | 33.6 | $6.5 \times 10^{-43}$ |

[1]Only 61 participants were included in this analysis (see text)

SCL-90 (Table 16). The size of the correlations with SCL-90, however, generally did not differ significantly between MACE and CTQ. The exceptions were the SCL-90 subscales Hostility and Phobic anxiety, both being significantly stronger associated with the MACE Sum-by-Duration score than with the CTQ Sum score. In addition, the MACE Sum-by-Duration score, compared with the CTQ Sum score, was nominally more strongly associated with the total scores as well as all nine subscale scores on the SCL-90. Likewise, as compared with the CTQ Multiplicity score, the MACE Multiplicity score was nominally more strongly associated with the total scores and eight out of nine subscale scores on the SCL-90.

## Discussion

In a combined participant group with psychiatric outpatients and employees, item to subscale and scaling analysis indicated that each of 10 MACE subscales could be captured with 4 to 8 items, together comprising a Norwegian MACE with 55 items. Subsequent tests in the

**Table 13. Pearson correlation coefficients between total scores on MACE and CTQ.**

| MACE total scores (mean; SD) | CTQ Sum r, [95% CI] | CTQ Multiplicity r, [95% CI] |
|---|---|---|
| MACE Sum (36.2; 19.9) | $r = .81$ [0.72-.87]) | $r = .78$ [0.70-.85]) |
| MACE Sum 6 subscales* (23.4; 14.2) | $r = .86$ [0.79-.91]) | $r = .83$ [0.76-.89]) |
| MACE Multiplicity (4.2; 2.7) | $r = .80$ [0.73-.87]) | $r = .79$ [0.71-.85]) |
| MACE Multiplicity 6 subscales* (2.9; 2.0) | $r = .83$ [0.77-.88]) | $r = .82$ [0.75-.87]) |
| MACE Duration (11.6; 6.5) | $r = .63$ [0.52-.73]) | $r = .62$ [0.50-.72]) |

* MACE Sum and Multiplicity scores that include only 6 of 10 subscales with themes that also are covered by the CTQ

**Table 14. Pearson correlation coefficients between matching subscales of MACE and CTQ.**

| MACE subscale | CTQ subscale | r, p [95% CI] |
|---|---|---|
| Parental verbal abuse | Emotional abuse | $r = .68$, $p < .001$, [.57-.77] |
| Emotional neglect | Emotional neglect | $r = .74$, $p < .001$, [.64-.82] |
| Physical neglect | Physical neglect | $r = .71$, $p < .001$, [.56-.82] |
| Parental non-verbal emotional abuse | Emotional abuse | $r = .63$, $p < .001$, [.48-.74] |
| | Emotional neglect | $r = .56$, $p < .001$, [.39-.70] |
| Parental physical abuse | Physical abuse | $r = .82$, $p < .001$, [.73-.89] |
| Sexual abuse | Sexual abuse | $r = .81$, $p < .001$, [.64-.91] |

outpatient group indicated good psychometric properties of the Norwegian MACE, as discussed below.

The 10 MACE subscales were generally consistent with respect to gender and age, with two exceptions. First, we saw a gender difference for the Physical neglect subscale, indicating different response patterns on single items for men and women. Second, the subscale for Parental physical abuse had an age effect, with a different item pattern for older and younger participants. These results indicate differences within the participant group in the way males and females were neglected or perceived neglect, and in the way older compared to younger participants had been physically abused (or perceived abuse) by parents during their upbringing. While our data cannot directly illuminate reasons for these variations, the age effect for parental physical abuse might reflect changes in child-rearing practices, with physical punishment being more common in Norway before the late 1970s.

The test-retest analysis indicated excellent stability over two weeks for both total scores and subscale scores on the Norwegian MACE-55 in psychiatric outpatients. Compared to a study by Teicher and Parigger [15] in normal healthy participants in the US, the test-retest reliability level in our outpatient group was slightly higher numerically for both the MACE Sum score (ICC = 0.96 vs. 0.91), MACE Multiplicity score (ICC = 0.94 vs 0.88), and most of the 10 subscale scores (ICC range = 0.82–0.95, mean = 0.89 vs range 0.63–0.90, mean = 0.78). This possibly slightly higher test-retest reliability in Norwegian outpatients than in normal healthy participants in the US study may reflect that the former had an overall higher level of MACE scores or that it was a more heterogeneous participant group, as well as that the time from test to retest was substantially shorter in our study—two weeks compared to 66 days on average in Teicher and Parigger's study.

**Table 15. The ability of MACE subscales to predict clinically relevant exposure levels on the CTQ subscales.**

| MACE subscale | CTQ subscale | ROC area [95% CI] | Best n items /cut-off* | Sensitivity/ specificity |
|---|---|---|---|---|
| Parental verbal abuse | Emotional abuse | 0.87, [0.80–0.94] | 3 | 0.87/0.71 |
| Emotional neglect | Emotional neglect | 0.82, [0.74–0.91] | 5 | 0.81/0.71 |
| Physical neglect | Physical neglect | 0.83, [0.74–0.92] | 2 | 0.92/0.66 |
| Parental non-verbal emotional abuse | Emotional abuse | 0.77, [0.68–0.87] | 2 | 0.85/0.61 |
| | Emotional neglect | 0.78, [0.69–0.87] | 2 | 0.83/0.62 |
| Parental physical abuse | Physical abuse | 0.91, [0.81–1.00] | 3 | 0.86/0.88 |
| Sexual abuse | Sexual abuse | 0.91, [0.84–0.98] | 1 | 0.96/0.77 |

* The number of items on the MACE subscales that gave the best sensitivity and specificity scores for clinical cut-off on CTQ (= MACE clinical cut-off scores). For the four MACE subscales MACE with no corresponding CTQ subscales, the following cut-offs were set: Witnessing interpersonal violence to parents– 3; Witnessing violence to siblings– 2; Peer verbal/ emotional abuse– 3; Peer physical bullying– 2.

**Table 16. Pearson correlations for total scores on MACE and CTQ with SCL-90 total scores.** Hotelling's t test for within sample comparisons (n = 87 outpatients).

| SCL-90 variables (mean; SD) | Sum scores | | | Multiplicity scores | |
|---|---|---|---|---|---|
| | MACE SxD[*] | MACE Sum | CTQ Severity | MACE Multi | CTQ Multi |
| GSI (1.43; 0.76) | r = 0.409 p<0.001 | r = 0.331 p = 0.002 | r = 0.320 p = 0.002 | r = 0.346 p = 0.001 | r = 0.279 p = 0.009 |
| PST (55.28; 20.79) | r = 0.337 p = 0.001 | r = 0.293 p = 0.006 | r = 0.274 p = 0.009 | r = 0.324 p = 0.002 | r = 0.252 p = 0.018 |
| Somatization[&] (1.59; 0.98) | r = 0.369 p<0.001 | r = 0.303 p = 0.004 | r = 0.300 p = 0.005 | r = 0.327 p = 0.002 | r = 0.294 p = 0.006 |
| Obsessive-compulsive (1.89; 0.98) | r = 0.288 p = 0.007 | r = 0.201 ns | r = 0.198 p = 0.066 | r = 0.207 p = 0.054 | r = 0.130 ns |
| Interpersonal sensitivity (1.62; 0.98) | r = 0.393 p<0.001 | r = 0.324 p = 0.002 | r = 0.310 p = 0.003 | r = 0.337 p = 0.001 | r = 0.266 p = 0.013 |
| Depression (1.94; 1.01) | r = 0.278 p = 0.009 | r = 0.203 p = 0.060 | r = 0.242 p = 0.024 | r = 0.212 p = 0.048 | r = 0.219 p = 0.042 |
| Anxiety (1.48; 0.93) | r = 0.394 p<0.001 | r = 0.324 p = 0.002 | r = 0.329 p = 0.002 | r = 0.351 p = 0.001 | r = 0.281 p = 0.008 |
| Hostility (0.59; 0.67) | r = 0.299[1] p = 0.005 | r = 0.274[2] p = 0.010 | r = 0.154 ns | r = 0.265[2] p = 0.013 | r = 0.133 ns |
| Phobic anxiety (1.10; 0.89) | r = 0.339[1] p = 0.001 | r = 0.258 p = 0.016 | r = 0.185 ns | r = 0.245 p = 0.022 | r = 0.174 ns |
| Paranoid ideation (1.00; 0.80) | r = 0.403 p<0.001 | r = 0.359 p = 0.001 | r = 0.339 p = 0.001 | r = 0.359 p = 0.001 | r = 0.296 p = 0.005 |
| Psychoticism (0.74; 0.66) | r = 0.427 p < .001 | r = 0.411 p<0.001 | r = 0.411 p<0.001 | r = 0.444 p<0.001 | r = 0.344 p = 0.001 |

[*]MACE SxD = combined Sum and Duration score, see Methods. The MACE score was significantly (p≤0.05) higher than the concomitant CTQ score in a [1]two-sided Z-test (Z ≥1.96) or in a [2]one-sided Z-test (Z ≥1.65). The associations with SCL-90 variables were compared for (i) each of MACE SxD and MACE sum with CTQ sum, and (ii) MACE multi with CTQ multi. GSI = Global severity index. PST = Positive symptom total. [&]Mean of participant average item scores for each SCL-90 subscale ("subscale GSI scores").

Regarding convergent validity between MACE and CTQ, our correlational results are generally consistent with those of Teicher and Parigger [15] and of Isele and coworkers [16] in normal healthy participant groups. Our correlations were somewhat higher for total Sum scores (our r = 0.81 vs r = 0.74 in Teicher and Parigger and r = 0.75 in Isele et al.) and for the subscales that were comparable in the two instruments (means across subscales, our study r = 0.84 vs r = 0.59 in Teicher and Parigger and r = 0.62 in Isele et al.). One possible reason for the stronger correspondence in our study is the generally higher level of adverse childhood experiences in psychiatric outpatients than in general population samples. Another facet of our findings was that, compared to the MACE Sum and Multiplicity scores, the MACE Duration score was more moderately associated with the CTQ total scores (r's = 0.62–0.63). This is in line with the view that the temporal components (timing and duration) of childhood adversities convey additional information to measures of degree or dose of exposure [20]. Consistent with this view is that we found the numerically strongest associations with psychiatric symptoms for the MACE total score that combined dose (Sum score) and duration of exposure (MACE Sum-by-Duration score). Further corroborating the convergent validity of the Norwegian MACE is that MACE scores could predict clinically relevant exposure levels on the CTQ subscales with acceptable levels of sensitivity and specificity.

The large population study by Teicher and Parigger [15] in the US applied variance decomposition analysis and found that MACE on average could account for twice the variance in psychiatric symptoms compared to the CTQ and ACE instruments, thereby confirming predictive validity. In their complementary correlation analysis, they found numerically stronger correlations for MACE than for the other two exposure measures with depression, suicidal ideation, anxiety, somatization symptoms, dissociation, and "limbic irritability". Similarly, we found a pattern of numerically higher correlations for the MACE total scores, when compared to CTQ total scores, with total levels of psychiatric symptoms and with most of the psychiatric symptom domains as measured with the SCL-90. This pattern was strongest for the MACE Sum-by-Duration score that combines dose and duration of exposure. The Sum-by-Duration score was nominally more strongly associated with the SCL-90 global scores and with each of

the nine psychiatric subdomains when compared to the CTQ total scores. For this MACE score, the degree of improvement over the CTQ in predicting psychiatric symptoms was comparable to that found for the MACE Sum score by Teicher and Parigger. In that study, correlations between MACE Sum scores and CTQ Severity scores for seven symptom domains were, on average, r = 0.35 and r = 0.29, respectively, while we found correlations for the MACE Sum-by-Duration score and CTQ Sum score with the SCL-90 GSI score at r = 0.41 and r = 0.32, respectively. Combined, these studies suggest that the MACE items and its' unique temporal features predict psychiatric symptoms better than the CTQ in both normal healthy people and patients with moderately severe mental health problems.

### Strengths and limitations

In light of the lack of instruments assessing adverse childhood experiences in a comprehensive way in Norway, the present study provides a brief but psychometrically sound version of the MACE to Norwegian clinicians and researchers. An advanced statistical method of item-response theory was used. Further, MACE is the first measure including unique temporal features in the assessment of adverse childhood experiences. Finally, the study was based on adult reports from both a clinical and general population. However, one limitation was that an only moderately large participant group was included, specifically for the validity tests based on 90 outpatients. This is likely to have restricted the possibility to achieve statistically significant results, most notably in tests of the relative predictive powers of MACE and CTQ in terms of psychiatric symptoms. The latter tests also could have been improved by including a more varied participant group in these analyses than only psychiatric outpatients.

### Conclusions

The MACE is a novel instrument freely available for clinical use and research that is suitable for investigating adverse relational experiences in childhood and adolescence and their role in subsequent health problems and other sequels. We found that a Norwegian version of MACE with 55-items, administered as a self-report measure, could be derived based on data from a composite participant group of employees and psychiatric outpatients. In the outpatient group, the Norwegian MACE had excellent test-retest reliability over two weeks and concurrent validity with the CTQ, and it numerically superseded the CTQ in predicting psychiatric symptoms captured by the SLC-90. This scale is a useful tool for research questions addressing sensitive periods for childhood adversities and associated health phenotypes. Copyright of the Norwegian 55-item version is with the authors. Any commercial use of this version is prohibited.

### Supporting information

**S1 File. Figures from Rasch analyses of MACE subscales.**
(DOCX)

**S2 File. The Norwegian MACE-55 instrument.**
(DOCX)

**S3 File. Introductory questions to MACE-55 instrument.**
(DOCX)

**S4 File. MACE-55 Norwegian SPSS data file setup.**
(SAV)

**S5 File. MACE-55 Norwegian SPSS syntax file.**
(SPS)

**S6 File. Fosse et al MACE study Norwegian data file.**
(SAV)

# Acknowledgments

We are grateful to: clinicians at Vestre Viken who collaborated with including their own patients as participants; outpatients and employees at Vestre Viken who joined as study participants; Per-Erik Holo for programming and administering MACE in the Confirmit poll system; office personnel at Asker and Ringerike district psychiatric centers for logistic support; and Hilde Nymoen for assistance in plotting.

# Author Contributions

**Conceptualization:** Roar Fosse, Dag Vegard Skjelstad, Inga Schalinski, Dorothea Thekkumthala, Thomas Elbert, Chris Margaret Aanondsen, Hanne Klæboe Greger, Thomas Jozefiak.

**Data curation:** Roar Fosse, Inga Schalinski.

**Formal analysis:** Roar Fosse, Inga Schalinski.

**Funding acquisition:** Thomas Jozefiak.

**Investigation:** Roar Fosse, Dag Vegard Skjelstad, Chris Margaret Aanondsen, Hanne Klæboe Greger, Thomas Jozefiak.

**Methodology:** Roar Fosse, Dag Vegard Skjelstad, Inga Schalinski, Dorothea Thekkumthala, Thomas Elbert, Chris Margaret Aanondsen, Hanne Klæboe Greger, Thomas Jozefiak.

**Project administration:** Roar Fosse, Dag Vegard Skjelstad, Thomas Jozefiak.

**Validation:** Dorothea Thekkumthala.

**Writing – original draft:** Roar Fosse, Inga Schalinski.

**Writing – review & editing:** Roar Fosse, Dag Vegard Skjelstad, Inga Schalinski, Dorothea Thekkumthala, Thomas Elbert, Chris Margaret Aanondsen, Hanne Klæboe Greger, Thomas Jozefiak.

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
