## [Decision Letter · Decision Letter 0]

16 Jan 2020

PONE-D-19-31983

Measuring childhood maltreatment: psychometric properties of the Norwegian version of the Maltreatment and Abuse Chronology of Exposure (MACE) scale

PLOS ONE

Dear Dr. Fosse,

I wish you a Happy New Year!

Thank you for submitting your manuscript to PLOS ONE. After careful consideration, we feel that it has merit but does not fully meet PLOS ONE’s publication criteria as it currently stands. Therefore, we invite you to submit a revised version of the manuscript that addresses the points raised during the review process.

As you can see from the reviewers comments, only minor modifications are required. Please make sure that your article confirms with the publication policies of PLOS ONE, in particular in regard to data availability to the community.

We would appreciate receiving your revised manuscript by Mar 01 2020 11:59PM. To enhance the reproducibility of your results, we recommend that if applicable you deposit your laboratory protocols in protocols.io, where a protocol can be assigned its own identifier (DOI) such that it can be cited independently in the future. For instructions see: http://journals.plos.org/plosone/s/submission-guidelines#loc-laboratory-protocols

We look forward to receiving your revised manuscript.

Kind regards,

Torsten Klengel, MD PhD

Academic Editor

PLOS ONE

Reviewers' comments:

Reviewer's Responses to Questions

**Comments to the Author**

1. Is the manuscript technically sound, and do the data support the conclusions?

Reviewer #1: Yes

Reviewer #2: Yes

2. Has the statistical analysis been performed appropriately and rigorously? 

Reviewer #1: Yes

Reviewer #2: Yes

3. Have the authors made all data underlying the findings in their manuscript fully available?

Reviewer #1: Yes

Reviewer #2: Yes

4. Is the manuscript presented in an intelligible fashion and written in standard English?

Reviewer #1: Yes

Reviewer #2: Yes

5. Review Comments to the Author

Reviewer #1: Dear Authors,

I would like to congratulate you on this effort. It is becoming rare to come across manuscripts which are so nicely and succinctly written, and statistically sound. While I very much like the manuscript, there is one question I would like to ask in relation to the manuscript. Although it comes across as such but its not clearly stated that the scale was developed as a self-report measure. I am sure you are aware that there is no consensus on whether self-report measures are superior to interview based measures or the other way around. For instance, studies suggest the most common type of misreporting of child abuse is underreporting – that is, individuals who experienced childhood abuse report not having been abused in childhood (Fergusson, Horwood, & Woodward, 2000; Hardt & Rutter, 2004; Widom & Shepard, 1996). Individuals abused in childhood may not disclose such a history for a variety of reasons. They may not divulge information on this sensitive topic to an interviewer they do not know or with whom they do not feel comfortable (Della Femina, Yeager, & Lewis, 1990).

Furthermore, a study by Negriff et al., reported that their results showed that an average of 48% of maltreatment found by the Maltreatment Case Record Abstraction Instrument (MCRAI) for each type of maltreatment were unique cases not captured by the Comprehensive Trauma Interview (CTI), whereas an average of 40% self-reported maltreatment (CTI) was not indicated by the MCRAI.

I am just curious that since the original MACE was developed as a self-report measure while the German MACE (KERF) was an interviewer based measure and although you used the German version for translation purposes, it looks like you decided to go with the self-report method. Your psychometrics align well with both the original version and the German version but I would like to know a little bit more about your chosen method.

Kind Regards

Reviewer #2: This is a very interesting article, well-written and methodologically well performed. The aim was to develop a Norwegian MACE and to investigate its psychometric properties. The authors did assess test-retest reliability of the resulting MACE version, convergent validity between MACE and the gold standard CTQ, and predictive validity of MACE compared to CTQ for patients’ psychiatric symptoms. Overall all analysis are adequate, however since the authors reduced the original scale from 75 to 55 itens, they built a new version of the instrument. Therefore I strongly suggest to called this new instrument brief MACE, or modified MACE or something that could identify that the authors are working with a modified instrument (including the title). The total score and the other scores will be different and even the direct comparison with the other versions would not be possible.

6. PLOS authors have the option to publish the peer review history of their article (what does this mean?). If published, this will include your full peer review and any attached files.

Reviewer #1: Yes: Alaptagin Khan

Reviewer #2: No

---

## [Author Response · Author response to Decision Letter 0]

29 Jan 2020

Response to reviewers

PONE-D-19-31983

Fosse et al: Measuring childhood maltreatment: psychometric properties of the Norwegian version of the Maltreatment and Abuse Chronology of Exposure (MACE) scale.

Dear Editor,

Below, please find our response to the comments from the reviewers.

Sincerely yours

Roar Fosse

Reviewer 1

R1.1 “Although it comes across as such but its not clearly stated that the scale was developed as a self-report measure.”

Author response: We have added specifications that we administered the instrument as a self-report measure. See p. 2 (line 30), p. 5 (line 100), p. 31 (line 516).

Reviewer 2

R2.1 “…since the authors reduced the original scale from 75 to 55 items, they built a new version of the instrument. Therefore I strongly suggest to called this new instrument brief MACE, or modified MACE or something that could identify that the authors are working with a modified instrument (including the title).”

Author response: We have discussed this issue thoroughly within our group, and we find it best to continue with the wording/ labeling for MACE instruments that were used by the original developers and in subsequent US/ German publications. Here, the complete, initial, 75-item experimental version of the Maltreatment and Abuse Chronology of Exposure Scale is denoted as “MACE-X”, and subsequent versions, extracted from the MACE-X, have simply been labeled “MACE”. Our 55-item version lies very close to the “main” American 52-item version of MACE, and by being slightly more extensive than this version, we cannot justifiably label it as “brief”. On this basis, to denote the Norwegian version of MACE that we have developed, we now consistently write “Norwegian MACE” or “Norwegian MACE-55”, and some places only “MACE-55” or “MACE” if the context makes it clear that we talk about the Norwegian version. Furthermore, we denote the original, complete, experimental 75-item version as “MACE-X”. In addition, we have made changes to the manuscript at places where we ourselves had used the term “brief” to denote the Norwegian MACE. See R2.1. on: p. 2 (lines 27-30), p. 4 (lines 77-78 and 88-94), p. 5 (lines 100-105, 116), p. 6 (line 122, 129), p. 7 (line 159), p. 10 (lines 229, 233), p. 11 (line 255), p. 22 (lines 381-382), p. 24 (line 396).

Other changes

We have changed the affiliation of our third author on the paper, Inga Schalinski, to: “Charité – Universitätsmedizin Berlin, corporate member of Freie Universität Berlin, Humboldt-Universität zu Berlin, and Berlin Institute of Health (BIH), Institute of Medical Psychology, Berlin, Germany”

---

## [Decision Letter · Decision Letter 1]

12 Feb 2020

Measuring childhood maltreatment: psychometric properties of the Norwegian version of the Maltreatment and Abuse Chronology of Exposure (MACE) scale

PONE-D-19-31983R1

Dear Dr. Fosse,

We are pleased to inform you that your manuscript has been judged scientifically suitable for publication and will be formally accepted for publication once it complies with all outstanding technical requirements.

As you can see from the comments below, reviewer #2 was not available at this point to re-assess your response to her/his comments. However, I believe that the minor comments raised during the first revision are adequately addressed and thus proceeded with my decision.

With kind regards,

Torsten Klengel, MD PhD

Academic Editor

PLOS ONE

Additional Editor Comments (optional):

Reviewers' comments:

Reviewer's Responses to Questions

**Comments to the Author**

1. If the authors have adequately addressed your comments raised in a previous round of review and you feel that this manuscript is now acceptable for publication, you may indicate that here to bypass the “Comments to the Author” section, enter your conflict of interest statement in the “Confidential to Editor” section, and submit your "Accept" recommendation.

Reviewer #1: All comments have been addressed

2. Is the manuscript technically sound, and do the data support the conclusions?

Reviewer #1: Yes

3. Has the statistical analysis been performed appropriately and rigorously? 

Reviewer #1: Yes

4. Have the authors made all data underlying the findings in their manuscript fully available?

Reviewer #1: Yes

5. Is the manuscript presented in an intelligible fashion and written in standard English?

Reviewer #1: Yes

6. Review Comments to the Author

Reviewer #1: Thank you for your response, and for making the necessary revisions. The manuscript looks set for publication.

7. PLOS authors have the option to publish the peer review history of their article (what does this mean?). If published, this will include your full peer review and any attached files.

Reviewer #1: Yes: Alaptagin Khan

---

## [Editor Report · Acceptance letter]

21 Feb 2020

PONE-D-19-31983R1 

Measuring childhood maltreatment: psychometric properties of the Norwegian version of the Maltreatment and Abuse Chronology of Exposure (MACE) scale 

Dear Dr. Fosse:

I am pleased to inform you that your manuscript has been deemed suitable for publication in PLOS ONE. Congratulations! Your manuscript is now with our production department. 

With kind regards,

on behalf of

Dr. Torsten Klengel 

Academic Editor

PLOS ONE